

# An enigmatic decoupling between heat stress and coral bleaching on the Great Barrier Reef

Thomas M. DeCarlo[1,2] and Hugo B. Harrison[3]

[1] Red Sea Research Center, Division of Biological and Environmental Science and Engineering, King Abdullah University of Science and Technology (KAUST), Thuwal, Saudi Arabia
[2] Australian Research Council Centre of Excellence for Coral Reef Studies, University of Western Australia, Crawley, WA, Australia
[3] Australian Research Council Centre of Excellence for Coral Reef Studies, James Cook University, Townsville, QLD, Australia

## ABSTRACT

Ocean warming threatens the functioning of coral reef ecosystems by inducing mass coral bleaching and mortality events. The link between temperature and coral bleaching is now well-established based on observations that mass bleaching events usually occur when seawater temperatures are anomalously high. However, times of high heat stress but without coral bleaching are equally important because they can inform an understanding of factors that regulate temperature-induced bleaching. Here, we investigate the absence of mass coral bleaching on the Great Barrier Reef (GBR) during austral summer 2004. Using four gridded sea surface temperature data products, validated with in situ temperature loggers, we demonstrate that the summer of 2004 was among the warmest summers of the satellite era (1982–2017) on the GBR. At least half of the GBR experienced temperatures that were high enough to initiate bleaching in other years, yet mass bleaching was not reported during 2004. The absence of bleaching is not fully explained by wind speed or cloud cover. Rather, 2004 is clearly differentiated from bleaching years by the slow speed of the East Australian Current (EAC) offshore of the GBR. An anomalously slow EAC during summer 2004 may have dampened the upwelling of nutrient-rich waters onto the GBR shelf, potentially mitigating bleaching due to the lower susceptibility of corals to heat stress in low-nutrient conditions. Although other factors such as irradiance or acclimatization may have played a role in the absence of mass bleaching, 2004 remains a key case study for demonstrating the dynamic nature of coral responses to marine heatwaves.

## INTRODUCTION

Mass coral bleaching events have increased in frequency since the 1980s in concert with rising sea surface temperature (SST) (*Donner et al., 2017*; *Hughes et al., 2018a*). Anomalously high summertime temperatures disrupt the symbiosis between coral host and zooxanthellae symbionts, leading to "bleaching" as the colorfully-pigmented

Corresponding author
Thomas M. DeCarlo, thomas.decarlo@kaust.edu.sa

symbionts are expelled and the coral skeleton is revealed (*Jokiel & Coles, 1977*; *Glynn, 1983*; *Gates, Baghdasarian & Muscatine, 1992*). The correspondence of mass bleaching events with anomalously high SST is often considered indicative of a direct relationship between heat stress and coral bleaching (*Glynn, 1993*; *Heron et al., 2016*; *Hughes et al., 2017*, *2018a*). However, the severity of mass bleaching events varies in space and time, and a range of environmental factors besides temperature can affect the bleaching sensitivity of corals. For example, changes in solar irradiance (*Brown, 1997*; *Dunne & Brown, 2001*), and nutrient concentrations (*Wooldridge, 2009*; *Cunning & Baker, 2012*; *Wiedenmann et al., 2013*; *Vega Thurber et al., 2014*) or ratios (*Wiedenmann et al., 2013*; *Morris et al., 2019*), can either exacerbate or ameliorate temperature stress to determine the severity and extent of bleaching.

Four mass bleaching events have been recorded on Australia's iconic Great Barrier Reef (GBR) and all have been associated with anomalously high SST. The summer of 1997–1998 was, at that time, the most extensive bleaching event observed on the GBR, with ~42% of reefs showing signs of bleaching (*Berkelmans & Oliver, 1999*). The GBR experienced another large-scale bleaching event just 4 years later in 2002, which affected ~54% of reefs (*Berkelmans et al., 2004*). In both cases, severe bleaching was predominantly observed on coastal reefs (*Berkelmans et al., 2004*). The most severe and widespread bleaching on the GBR occurred in 2016, when comparative surveys suggest approximately 91% of reefs showed signs of bleaching, with severe bleaching leading to widespread mortality in the northern third of the GBR (*Hughes et al., 2017*). This was followed by the first recorded back-to-back bleaching in 2017, which affected at least one third of the GBR and the remote atolls in the Coral Sea (*Hughes & Kerry, 2017*; *Harrison et al., 2018*). In each case, the geographic footprint of bleaching on the GBR was strongly correlated with the intensity and duration of SST anomalies (*Hughes et al., 2017*), a pattern that mirrors the response of corals to SST anomalies globally (*Hughes et al., 2018a*). It is also important to note that extensive bleaching and mortality was observed in 1982 at the majority of reefs visited by researchers (*Harriott, 1985*; *Oliver, 1985*), yet these observations were restricted to just 14 reefs in the central GBR and Lizard Island in the north, making it difficult to distinguish whether 1982 was a minor or major bleaching event.

Observations of bleaching events have informed a strong causal link between SST anomalies and the severity of bleaching (*Glynn, 1983*; *Bruno et al., 2001*; *Shuail et al., 2016*; *Barkley et al., 2018*; *Hughes et al., 2018b*), but heat stress has not always led to bleaching (*Guest et al., 2012*; *Pratchett et al., 2013*). Assessments of coral bleaching sensitivities have typically been retrospective; identifying the conditions that were necessary to trigger bleaching. This has led to clearly defined temperature thresholds above which bleaching is expected (*Glynn, 1983*; *Shuail et al., 2016*; *Hughes et al., 2017*). However, it is critical to consider periods when these thresholds were crossed, but bleaching did not occur. Such cases are important because they can inform our understanding of coral tolerance to heat stress, thereby assisting efforts to predict when and where to find the most resilient coral reef ecosystems (*Beyer et al., 2018*; *Hoegh-Guldberg et al., 2018*). Indeed, case studies documenting the absence of bleaching under anomalously high summer SST have attributed the corals' tolerance to acclimatization (*Guest et al., 2012*; *Gintert et al., 2018*),

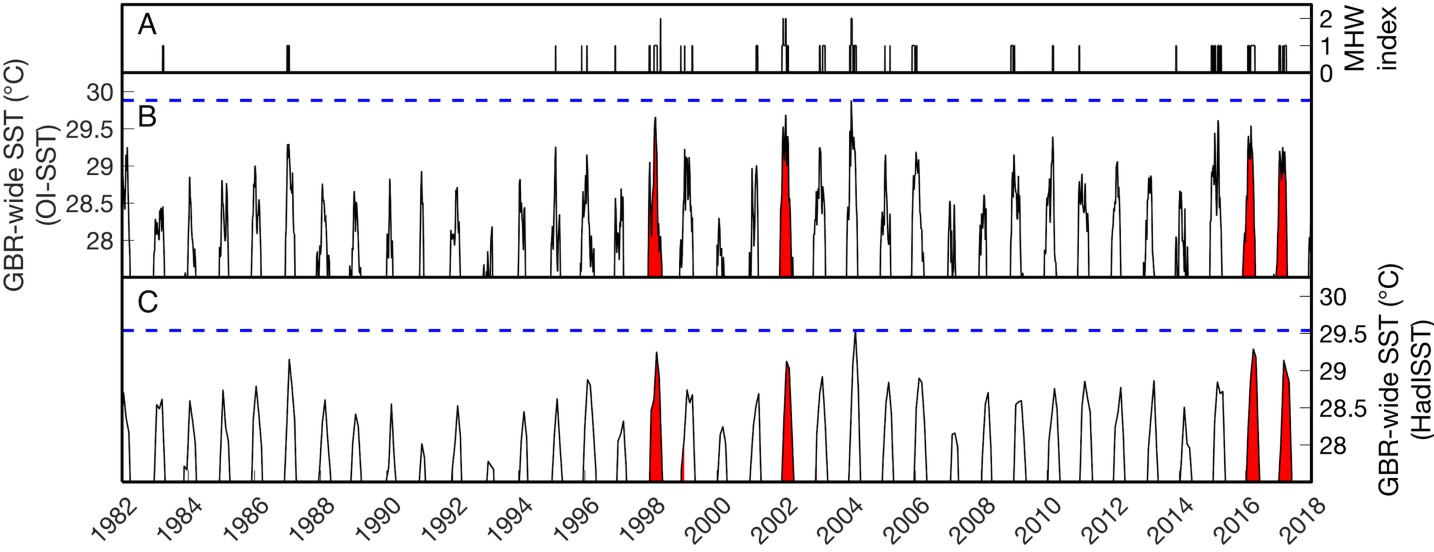

**Figure 1 Sea surface temperature (SST) time series of the Great Barrier Reef (GBR).** (B–C) show the average SST of all grid boxes in the weekly NOAA OI-SSTv1 or monthly HadISST dataset covering the entire GBR from 1982 to 2017. The vertical axes are restricted to >27.5 °C to highlight summertime temperatures. Red background coloring indicates the years of severe mass coral bleaching reported in the GBR (1998, 2002, 2016, and 2017). The horizontal blue dashed lines indicate the maximum GBR-wide SST in each dataset. On average across the GBR, the highest temperatures of the satellite era were reached in 2004 (note that this does not mean 2004 was the warmest summer everywhere on the GBR; see spatial analyses below). (A) shows the summertime Marine Heatwave (MHW) index as defined by *Hobday et al. (2018)*. Category 2 ("Strong") heatwaves occurred GBR-wide in 1998, 2002, and 2004.

adaptation (*Osman et al., 2018*), high frequency temperature variability (*Safaie et al., 2018*), night-time cooling (*Richards et al., 2019*), cloud cover (*Skirving et al., 2017*), or unique environmental settings such as exposure to internal waves (*Schmidt et al., 2016*; *DeCarlo et al., 2017*; *Green et al., 2019*; *Reid et al., 2019*).

Despite the importance of times with high temperatures but not coral bleaching, such events can be difficult to identify due to reporting bias. Bleaching databases that document the occurrence of bleaching rather than absence thereof are of limited utility because they rarely distinguish between the absence of bleaching and the absence of observations (*Donner et al., 2017*; *Oliver, Berkelmans & Eakin, 2018*). We are therefore limited to case studies where bleaching absence was either directly reported (*Guest et al., 2012*; *Pratchett et al., 2013*; *DeCarlo et al., 2017*; *Gintert et al., 2018*) or to locations with continuous long-term monitoring programs such as the GBR.

Here, we investigate the conspicuous absence of severe coral bleaching on the GBR during the austral summer of 2004. This was one of the warmer summers–and perhaps *the warmest summer*—of the satellite era (1982–2017) on the GBR (Fig. 1), but widespread or severe bleaching was not reported. We first use a suite of 52 in situ temperature loggers deployed across the GBR in 2004 to validate satellite-derived SST for capturing reef-water temperatures, and to identify which of four satellite SST products best represents 2004 GBR heat stress. We then investigate potential reasons for the absence of bleaching reports during 2004, including the possibility that widespread bleaching occurred but went unnoticed, that some environmental factor mitigated heat stress, or that corals acclimatized following previous bleaching events in 1998 and 2002.

## MATERIALS AND METHODS

### Data sources of satellite-derived sea surface temperatures

In situ temperature logger data were acquired from the *Australian Institute of Marine Science (AIMS) (2018)* for all available GBR and Coral Sea sites during the austral summer of 2004 ($n$ = 52 loggers with continuous measurements throughout the summer). Additionally, temperature data for mass bleaching events on the GBR (1998, 2002, 2016, 2017; see *Berkelmans & Oliver, 1999*; *Berkelmans et al., 2004*; *Hughes et al., 2018a*; *Oliver, Berkelmans & Eakin, 2018*) were downloaded for all loggers deployed in both 2004 and at least one bleaching year. Temporal resolutions typically ranged from 10 to 30 mins, and nominal water depths varied from 0 to 20 m. We note that nominal depths of zero m indicate deployment within a few meters of the surface, although the specific depth was not recorded (Australian Institute of Marine Science, 2018, personal communications).

Gridded SST data were acquired for heat stress analyses from four sources: the National Oceanic and Atmospheric Administration (NOAA) Coral Reef Watch (CRW) "CoralTemp" version 1.0 (*NOAA Coral Reef Watch, 2019*; *Liu et al., 2014*; *Heron et al., 2016*), Optimum Interpolation SST (OI-SSTv1) (*Reynolds et al., 2002*), the higher-resolution OI-SSTv2 that is based primarily on the advanced very high resolution radiometer (AVHRR) satellite (*Reynolds et al., 2007*; *Banzon et al., 2016*), and the *Canadian Meteorological Center (CMC) (2012)*. It is important to note that these SST products are not entirely independent because they share some of the same data (*Liu et al., 2014*), but they nevertheless serve as alternative estimates of SST that can be validated with in situ measurements. We refer to these as satellite-SST products, although the OI products blend measurements from satellites, ships, and buoys (*Reynolds et al., 2002*; *Banzon et al., 2016*). Both OI-SSTv1 and OI-SSTv2 cover the period 1982–2017, whereas CRW covers 1985–2017, and CMC covers 1993–2017. The temporal resolution is daily for CRW, OI-SSTv2, and CMC, and weekly for OI-SSTv1. CRW has the highest spatial resolution at 0.05° (approximately five km), OI-SSTv2 is at 0.25°, CMC is at 0.2°, and OI-SSTv1 is at 1° resolution. We re-gridded CMC, OI-SSTv1, and OI-SSTv2 to the same 0.05° grid used by CRW with simple linear interpolation. This enabled direct pixel-by-pixel comparisons among the SST products, even though the true resolution of these products were coarser than the 0.05° grid.

An additional analysis was conducted on SST data from the HadISST product (*Rayner et al., 2003*). However, since HadISST is resolved only monthly, it is poorly suited for the calculation of summertime heat stress, which often arises from anomalously warm days or weeks. Nevertheless, HadISST allows a separate analysis of SST variability over time for comparison to the four main SST products used in this study.

### Data sources of other environmental variables

Outgoing longwave radiation (OLR) data were acquired from the National Center for Atmospheric Research (NCAR) (*Liebmann & Smith, 1996*). Surface wind data were acquired from the National Center for Environmental Prediction/NCAR Reanalysis (*Trenberth & Olson, 1988*; *Kalnay et al., 1996*). Both OLR and wind datasets are of daily

temporal resolution and 2.5° spatial resolution, which is much coarser than the scale of individual coral reefs. Photosynthetically active radiation (PAR) data (*Frouin, Franz & Werdell, 2002*) were acquired from the moderate resolution imaging spectroradiometer (MODIS) Terra (2002–2017) and SeaWiFS (1998) satellites as 8-day composites at nine-km resolution.

Ocean currents and temperature, sea surface height (SSH), and surface wind stress were acquired from the Simple Ocean Data Assimilation (SODA) version 3.3.1 at monthly temporal resolution and 0.25° spatial resolution (*Carton et al., 2018*). SODA is a state-of-the-art ocean modeling system constrained by observations when and where they are available, and it is well-suited for quantifying temporal changes in sea level and ocean circulation in the Coral Sea and along the Australian continent (*Chepurin, Carton & Leuliette, 2014*; *Zhai et al., 2014*).

Sub-daily, high-resolution (0.25°) wind speeds during austral summer 2004 were acquired from QuikSCAT (*Perry, 2001*). These data were used to assess the wind fields during major cyclones near the GBR or Coral Sea.

We used the World Atlas of Coral Reefs for locations of the reefs of the GBR, downloaded from ReefBase (*Spalding, Ravilious & Green, 2001*).

## Calculation of degree heating weeks

We calculated degree heating weeks (DHW) to assess heat stress using both in situ loggers and satellite SST products. First, we calculated "Hotspots" as:

$$\text{Hotspots} = \text{Temperature} - \text{MMM} \tag{1}$$

where temperature is the daily (CRW, CMC, and OI-SSTv2), weekly (OI-SSTv1), or monthly (HadISST) mean temperature and maximum monthly mean (MMM). For consistency, we used the MMM climatology from NOAA CRW CoralTemps when calculating Hotspots from all temperature sources. We assessed whether this approach adds bias to OI-SSTv2 heat stress metrics by comparing the climatologies calculated from CRW and OI-SSTv2 (Figs. S1 and S2). However, we found there was not a systematic offset between the two climatologies, meaning that the OI-SSTv2 Hotspots calculated with the CRW climatology are not substantially influenced by this approach (Fig. S1). Furthermore, the differences between the products vary strongly in time (Fig. S2), which cannot arise from the use of a single climatology. The MMM climatology is based on average monthly temperatures during 1985–2012 using AVHRR data (*Liu et al., 2006*, *2014*). Next, we calculated DHW as:

$$\text{DHW} = \frac{\sum_{1}^{84 \text{ days}} \text{Hotspots (if Hotspots} > 1)}{7 \text{ days}} \tag{2}$$

where only Hotspots exceeding 1 °C above the MMM contribute to DHW, and the sum of Hotspots during the previous 84 days (12 weeks) contribute to DHW.

## Comparison of in situ and satellite-derived SST

We compared daily (or weekly for OI-SSTv1) mean temperature between in situ loggers and the four satellite SST products for the years 1998, 2002, 2004, 2016, and 2017. Initially,

this analysis included all available measurements of daily (or weekly) in situ temperature from anywhere on the GBR or in the Coral Sea between October of the preceding year through June of each year. We extracted SST from the satellite products for the grid cell encompassing each logger, and temporally matched the in situ and satellite time series to enable direct comparisons. From these spatially and temporally matched daily temperatures, we calculated root mean square errors (RMSE) and biases (average of satellite—logger) for each year and each satellite product, separately. We repeated these calculations for only Hotspot times because these are the only data contributing to the DHW metric commonly used in coral bleaching assessments.

Additionally, we compared the heat stress calculated between loggers and satellite products. For this analysis, we used only those loggers with continuous data from the start of January through the end of March of each year. We calculated the mean number of hotspot days, total hotspots, and maximum DHW for each logger and satellite product, separately for each year.

## Assessment of other environmental variables

We considered whether some other environmental factor besides temperature may have played a role in bleaching susceptibility during 2004. OLR was used as a proxy measure of light stress because OLR is affected by convective cloudiness (*Reed, 1976*; *Kessler & Kleeman, 2000*), which in turn is expected to influence the amount of sunlight reaching the sea surface. We recognize that OLR is an imperfect proxy for irradiance reaching coral colonies because irradiance may also be affected by turbidity, water depth, surface waves, and different types of clouds (*Skirving et al., 2017*). Nevertheless, since OLR is available in a daily gridded product over the past several decades, it is the most feasible way to compare irradiance on the GBR between years ranging from 1998 to 2017. The PAR data were also used to assess radiative stress, although measurements from the Terra satellite only began in 2000. Additionally, we assessed temporal changes in wind speed because low-wind conditions could reduce wave heights and modulate the transmission of solar irradiance across the sea surface (*Payne, 1972*; *Stramska & Dickey, 1998*). For both the OLR and wind-speed datasets, we first calculated a weekly climatology (using a base period of 1998–2016), and then compared the daily anomalies around this climatology between 2004 and bleaching years. For PAR, we calculated March anomalies relative to the 2000–2017 March climatology because this is typically the month when the highest DHW occurs on the GBR.

Furthermore, we assessed the oceanographic conditions surrounding the GBR during 1998, 2002, and 2004 using SODA (note that the years 2016 and 2017 are not included in the most recent version of SODA). Specifically, we compared the velocity of the East Australian Current (EAC), SSH, tilting of isotherms, and wind-driven upwelling between these 3 years. The EAC is the poleward-flowing western boundary current extending southeastward from the bifurcation point of the South Equatorial Current (SEC) colliding with the Australian continental shelf at around 15° S, near the central to northern GBR. First, we calculated monthly anomalies in near-surface velocity (0–25 m), SSH, and

SST; and plotted maps of these anomalies during austral summers (January, February, March; or "JFM"). We also calculated wind stress curl as:

$$\text{Curl} = \nabla = \frac{\partial \tau_y}{\partial_x} - \frac{\partial \tau_x}{\partial_y} \qquad (3)$$

where $\tau$ is wind stress at the sea surface, which is an output provided directly by SODA, and $x$ and $y$ are eastward and northward, respectively. Ekman vertical transport ($\omega_E$) was then calculated from wind stress curl as:

$$\omega_E = \frac{\nabla}{\rho f} \qquad (4)$$

where $\rho$ is seawater density, $f$ is the Coriolis frequency, and where positive (negative) $\omega_E$ indicates upwelling (downwelling). Finally, we calculated and plotted maps of $\omega_E$ anomalies during JFM of 1998, 2002, and 2004.

## RESULTS

### Satellite-derived heat stress on the Great Barrier Reef

The level of heat stress (maximum annual DHW) on the GBR between 1998 and 2017 varies dramatically between the four SST products (Fig. 2). For example, in 1998, CRW implies that DHWs across nearly the whole GBR did not exceed 2–3 °C-weeks, whereas OI-SSTv1 implies that large swaths of the southern and far northern GBR were exposed to 7–8 °C-weeks. DHWs derived from CRW during both 2002 and 2004 were high in the Coral Sea but nearly the entire GBR remained less than 3 °C-weeks. Conversely, OI-SSTv2 and CMC show that the high DHWs in the Coral Sea during 2002 and 2004 extended all the way to the Australian coastline, with most of the GBR experiencing 5–10 °C-weeks during these 2 years. OI-SSTv1 implies even greater heat stress during 2002 and 2004, with sections of the southern and northern GBR reaching 12–15 °C-weeks. The spatial patterns of DHW in 2016 and 2017 are very similar among CRW, OI-SSTv2, and CMC, but CRW shows DHWs several °C-weeks higher than OI-SSTv2 throughout much of the central and northern GBR. Similar patterns exist in annual maximum SST anomalies and in HadISST (Figs. S3–S4).

Due to these differences in DHW histories, the year of maximum DHW varies immensely between the four satellite SST products (Fig. 3). According to CRW, either 2016 or 2017 was the year of greatest DHW on more than 60% of the reefs of the GBR, with only a portion of the far southern GBR undergoing maximum DHW in 2002. In contrast, OI-SSTv1, OI-SSTv2, and CMC imply that less than 35% of reefs experienced their maximum DHW in 2016 or 2017, and instead that either 2002 or 2004 was the year of highest DHW on nearly the entire southern and central sections of the GBR.

### Comparison of in situ temperature loggers and satellite-derived SST

Scatterplots of temperature measured by in situ loggers and satellites are shown for each year and each SST product in Fig. 4, and metrics of comparison are listed in Tables 1 and 2.

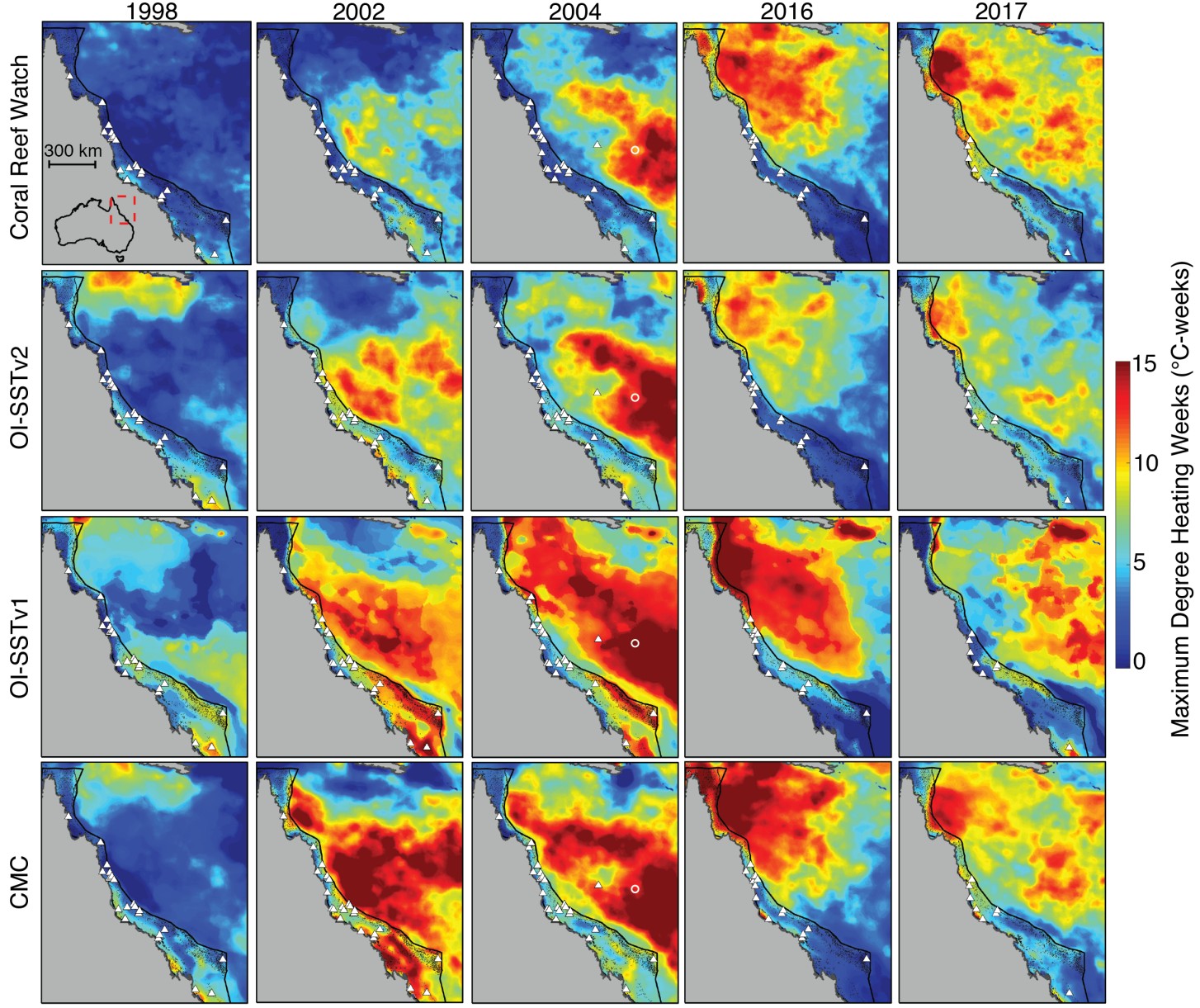

**Figure 2 Heat stress on the GBR during austral summers of 2004 and years when mass coral bleaching was observed.** Rows correspond to different satellite-SST products and columns show the key years assessed in this study. Colors represent the maximum DHW per five-km pixel during each year. White triangles indicate the locations of in situ temperature loggers used each year in the validation of satellite-derived SST. Black dots represent the reefs of the GBR and Coral Sea. The white circle in 2004 maps indicates Lihou Reef, where 65% bleaching was observed.

In general, CRW and CMC were the most precise SST products (lowest RMSEs), and the precision of each SST product increased over time. However, despite the relatively good precision of CRW, it was not the most accurate (lowest bias) for 1998, 2002, and 2004. Rather, CRW consistently underestimated in situ temperature during these 3 years, particularly during the warmest days of the year (hotspot day biases of −0.74 to −0.42 °C). OI-SSTv2 was overall the most accurate SST product during 1998, 2002, and 2004, and

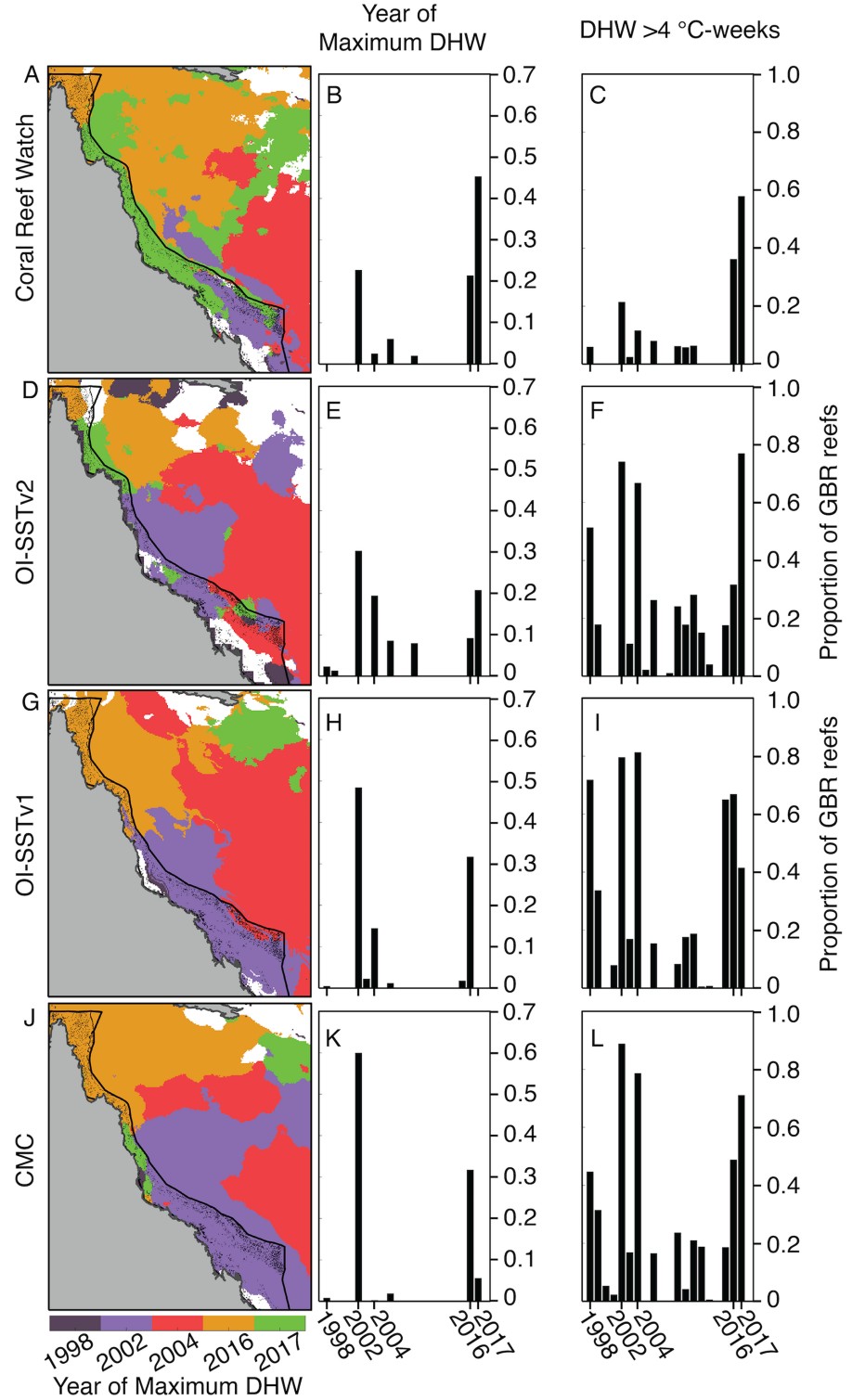

**Figure 3 Year of maximum DHW (1998–2017).** In (A, D, G, J), colors indicate the year with highest DHW. Locations where the highest DHW occurred in any year other than 1998, 2002, 2004, 2016, or 2017 are colored white. Histograms show the proportion of reefs on the GBR experiencing their highest DHW in each year (B, E, H, K), and the proportion of reefs with DHW exceeding 4 °C-weeks (C, F, I, L).

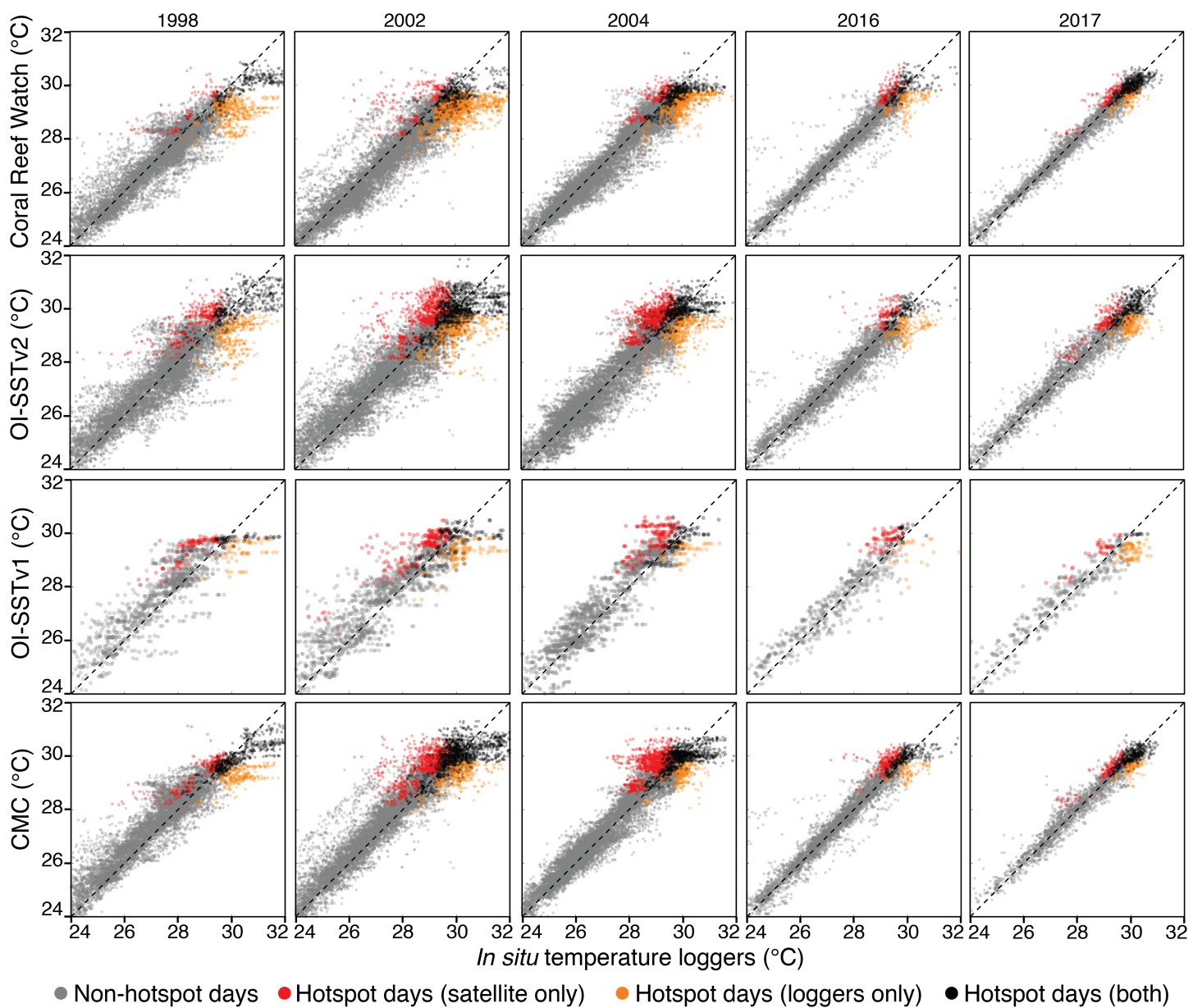

**Figure 4** **Satellite-logger temperature validations per year and per SST product.** The data are daily for CRW, OI-SSTv2, and CMC, and weekly for OI-SSTv1. Gray points are non-hotspot times, orange points are hotspot times only in loggers, red points are hotspot times only in satellite products, and black points are hotspot times in both the loggers and satellites. The dashed black lines indicate 1:1 relationships.

performed well at the highest temperatures (hotspot day biases of −0.20 to 0.12 °C). While the bias of OI-SSTv2 remained similar for 2016 and 2017 (hotspot day biases of −0.17 and −0.08 °C, respectively), CRW improved substantially relative to previous years (2016 and 2017 hotspot day biases of −0.15 and −0.08 °C, respectively).

Metrics of heat stress followed similar patterns to the biases in hotspot day temperatures (Table 2). During 1998, 2002, and 2004, the total number of hotspot days, the total hotspots (°C), and the maximum DHW recorded by in situ loggers most closely matched

**Table 1 Comparison of satellite-derived SST and in situ temperature measurements on the GBR and in the Coral Sea.**

| | 1998 | | | | 2002 | | | | 2004 | | | |
|---|---|---|---|---|---|---|---|---|---|---|---|---|
| | CRW | OI-SSTv2 | OI-SSTv1 | CMC | CRW | OI-SSTv2 | OI-SSTv1 | CMC | CRW | OI-SSTv2 | OI-SSTv1 | CMC |
| RMSE | 0.63 | 0.71 | 0.80 | **0.61** | 0.71 | 0.74 | 0.79 | **0.63** | **0.46** | 0.57 | 0.63 | 0.48 |
| Bias | −0.10 | **0.06** | 0.31 | 0.23 | −0.27 | **0.09** | 0.23 | 0.21 | −0.09 | **0.04** | 0.25 | 0.14 |
| RMSE (hotspots) | 1.12 | 1.00 | 1.03 | **0.80** | 1.16 | 0.96 | 0.97 | **0.84** | 0.75 | 0.80 | 1.11 | **0.74** |
| Bias (hotspots) | −0.74 | −0.20 | **0.02** | −0.30 | −0.73 | **−0.05** | 0.07 | 0.07 | −0.42 | **0.12** | 0.40 | 0.25 |
| | **2016** | | | | **2017** | | | | | | | |
| RMSE | **0.40** | 0.46 | 0.53 | **0.40** | **0.30** | 0.43 | 0.54 | 0.34 | | | | |
| Bias | **0.03** | −0.06 | 0.11 | 0.08 | **−0.01** | 0.02 | **0.01** | **0.01** | | | | |
| RMSE (hotspots) | 0.61 | 0.75 | 0.74 | **0.57** | **0.36** | 0.54 | 0.68 | 0.37 | | | | |
| Bias (hotspots) | −0.15 | −0.17 | 0.20 | **0.08** | −0.08 | −0.08 | 0.23 | **−0.04** | | | | |

Note:
CRW and OI-SSTv2 data are daily, and OI-SSTv1 is weekly. RMSE indicates the root mean square error between daily or weekly satellite and logger temperature data (°C). Bias indicates the mean difference between satellite and logger temperature data (°C). Hotspots refers to the days (or weeks for OI-SST) with temperature >1 °C above the maximum monthly mean (MMM), and thus contribute to degree heating weeks (DHW). For each metric and each year, the best performing satellite product is in bold and underlined.

**Table 2 Comparison of satellite-derived and in situ logger estimates of heat stress metrics.**

| | 1998 | | | | | 2002 | | | | | 2004 | | | | |
|---|---|---|---|---|---|---|---|---|---|---|---|---|---|---|---|
| | CRW | OI-SSTv2 | OI-SSTv1 | CMC | Loggers | CRW | OI-SSTv2 | OI-SSTv1 | CMC | Loggers | CRW | OI-SSTv2 | OI-SSTv1 | CMC | Loggers |
| Hotspot days | 10.5 | **20.8** | 25.1 | 20.0 | 21.2 | 14.7 | 41.7 | **40.1** | 46.5 | 38.5 | 13.1 | **31.4** | 31.6 | 37.7 | 23.1 |
| Total hotspots | 14.3 | 31.2 | **31.7** | 27.4 | 35.1 | 18.6 | **61.2** | 53.3 | 68.7 | 62.7 | 16.4 | **42.4** | 43.4 | 50.3 | 33.5 |
| Maximum DHW | 2.0 | 4.4 | **4.5** | 3.8 | 4.9 | 2.4 | 7.8 | **7.6** | 8.9 | 7.6 | 2.2 | **5.5** | 6.2 | 6.7 | 4.5 |
| | **2016** | | | | | **2017** | | | | | | | | | |
| Hotspot days | **14.4** | 12.8 | 24.7 | 22.0 | 17.5 | **40.4** | 35.0 | 21.7 | 33.5 | 42.0 | | | | | |
| Total hotspots | 18.7 | 16.7 | 31.0 | **28.3** | 23.9 | **51.0** | 46.9 | 25.4 | 42.1 | 55.4 | | | | | |
| Maximum DHW | **2.7** | 2.4 | 4.4 | 4.0 | 3.3 | **7.3** | 6.5 | 3.6 | 6.0 | 7.5 | | | | | |

Notes:
Hotspot days are the total number of days with temperature >1 °C above the maximum monthly mean (MMM). Total hotspots are the number of hotspot days multiplied by average anomaly relative to the MMM for those days. Degree heating weeks (DHW) are defined in the text, and integrate both the magnitude and duration of temperatures anomalies exceeding the MMM. All three metrics are presented as the average among available loggers (and the grid boxes covering those logger sites). For each metric and each year, the best performing satellite product is in bold and underlined.
Note that the loggers do not represent a random sampling of GBR locations, and the locations of loggers were not the same across years, meaning that the absolute values of heat stress metrics presented here should be compared within each year but not between years.

either OI-SSTv1 or OI-SSTv2, with CRW underestimating each metric. Conversely, in 2016 and 2017, CRW most closely matched the loggers for every metric except that total hotspots during 2016 were best represented by CMC.

The agreement in maximum DHW between in situ loggers and satellite SST products during 2004 varied spatially (Fig. 5). OI-SSTv2 best captured maximum DHW at Coringa-Herald, the only logger site in the Coral Sea, which experienced among the highest DHWs during 2004 (DHWs of 11, 9, and 5 °C-weeks in the logger, OI-SSTv2, and CRW, respectively). Additionally, OI-SSTv2 performed best for most sites in the northern and north-central GBR, and for most inshore reefs. However, maximum DHWs of a cluster of reefs in the central GBR were best represented by CRW. Thus, while DHWs in

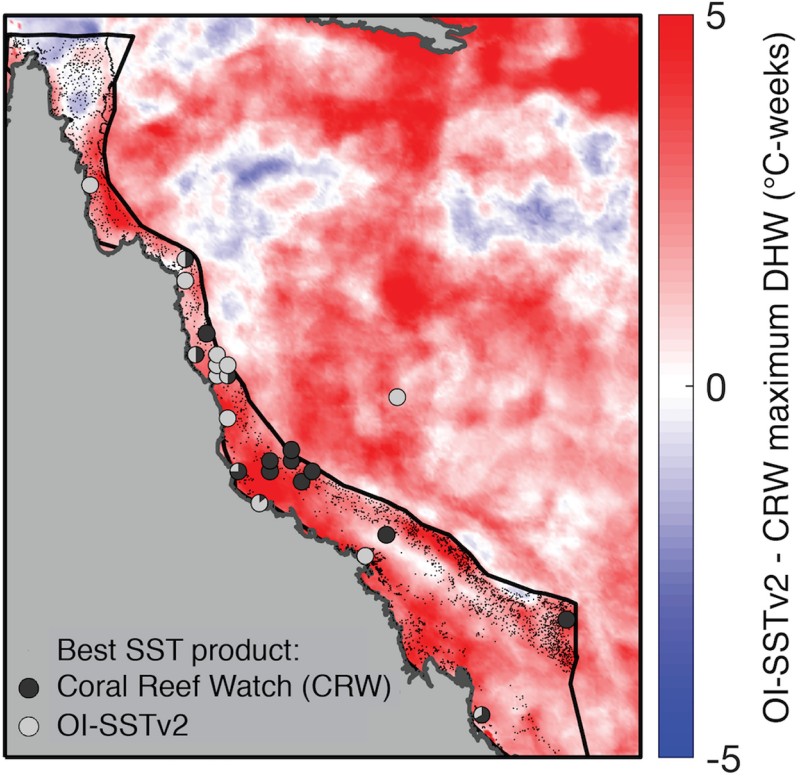

**Figure 5 Comparison of DHW derived from CRW and OI-SSTv2 during 2004.** Colors show the difference in maximum DHW, where red indicates that OI-SSTv2 estimates higher DHW than CRW. Circles show where temperature loggers best matched maximum DHW in CRW (dark gray) or OI-SSTv2 (light gray).

OI-SSTv2 were higher than CRW for nearly the entire GBR (Fig. 5), the reality was likely somewhere in between the two but closer overall to OI-SSTv2 (e.g., metrics in Tables 1 and 2). There was a significant depth-dependence of the difference in maximum annual DHW between OI-SSTv2 and loggers ($p = 0.019$), but it explained relatively little variance ($r^2 = 0.03$) (Fig. S5).

## Irradiance, winds, and currents

Summertime (JFM) OLR anomalies were generally positive, on average, for 2004 and bleaching years. Mean (±1 standard deviation) JFM daily OLR anomalies were 3 ± 23, 14 ± 19, 1 ± 24, 13 ± 28, and −1 ± 25 W m$^{-2}$ for 1998, 2002, 2004, 2016, and 2017, respectively. Thus, while 2004 was characterized by lower OLR (and by inference, lower irradiance) than 2002 and 2016, the large variance relative to the average differences makes it difficult to conclude that there were meaningful disparities between these years (Fig. 6; see also Figs. S9–S11 for regional analyses). The 8-day PAR composites shown in Fig. 6 track the OLR anomalies closely, indicating that OLR is indeed a reasonably effective proxy for cloud cover. However, it is also important to recognize that the seasonal cycle of PAR (e.g., peaking at the solstice in December and declining through the summer on the GBR) is not represented by OLR anomalies. March PAR anomalies were

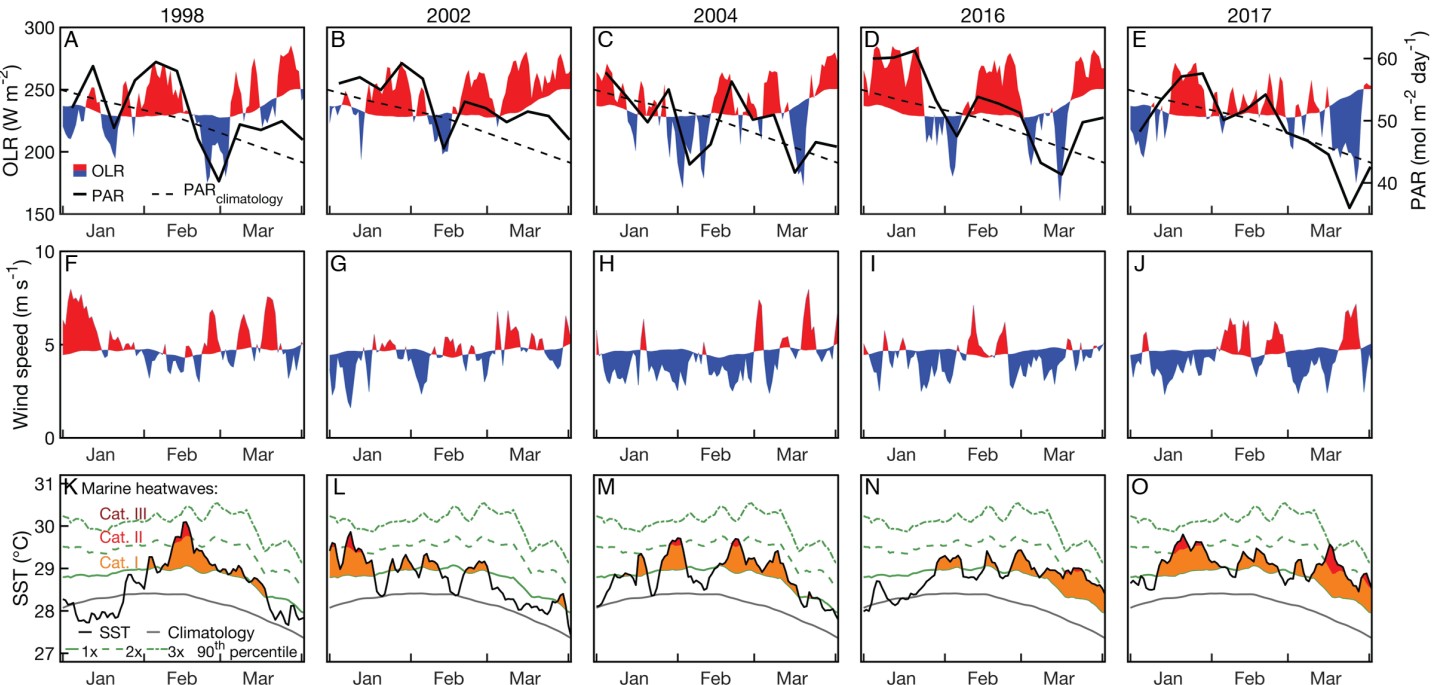

**Figure 6 Outgoing longwave radiation (OLR) and photosynthetically active radiation (PAR; A–E), surface wind speed (F–J), and SST (OI-SSTv2; K–O) time series during January, February, and March of 2004 and years when bleaching was observed.** Red indicates anomalously high and blues indicate anomalously low OLR or winds relative to the weekly climatology. Only grid boxes covering GBR reefs were included in the analysis. The SST panels also indicate marine heatwave categories, as defined by *Hobday et al. (2018)*.

highest overall in 2002 when almost the entire GBR was exposed to unusually high PAR, whereas 2004, 2016, and 2017 were characterized by a mixture of negative and positive anomalies (Fig. 7). The spatial patterns of positive PAR anomalies during 2016 and 2017 were broadly similar to that of bleaching, being mostly concentrated in the northern GBR. During 2004, similar or higher numbers of reefs experienced positive PAR anomalies as 2016 and 2017, and the highest PAR values were in the central to southern GBR, consistent with where the highest DHW occurred (Fig. 2).

Wind speed anomalies tended to be negative during 2004 and bleaching years (Fig. 6). Mean JFM daily wind speed anomalies were $0.5 \pm 1.2$, $0.3 \pm 1.0$, $-0.5 \pm 1.1$, $-0.4 \pm 0.9$, and $-0.2 \pm 1.2$ m s$^{-1}$ for 1998, 2002, 2004, 2016, and 2017, respectively.

The timing of maximum SST, and the temporal difference between SST and PAR maxima, did not clearly distinguish 2004 from bleaching years. With the exception of 2002, when SST maxima occurred relatively early in the season, the other years are generally characterized by SST maxima between the 30th and 60th days of the year, which are usually 40–80 days after peak PAR (Fig. 8). We also do not find a clear pattern between the alignment of temperature and PAR peaks and the onset of bleaching. For example, 1998 SST peaked in mid-February, shortly after large OLR and PAR anomalies in early February (Fig. 6). However, 2002 SSTs were highest in early January, during a time of near-climatological OLR and PAR. The late-January SST peak during 2004 occurred under

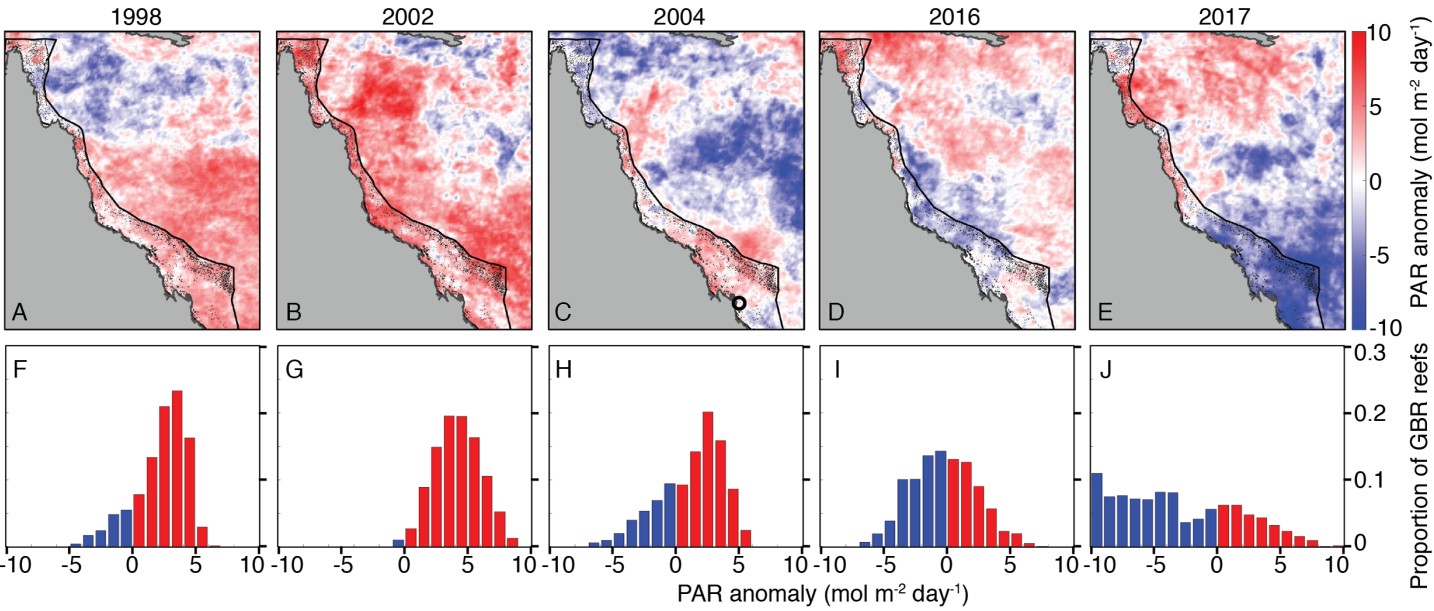

**Figure 7 Photosynthetically active radiation (PAR) March anomalies during 1998, 2002, 2004, 2016, and 2017 (A–E).** Histograms (F–J) show the proportions of GBR reefs with positive (red) and negative (blue) PAR anomalies in March of each year. The black circle in the 2004 map indicates the location of the Keppel Islands, where *Skirving et al. (2017)* assessed PAR and heat stress.

slightly positive PAR anomalies, but was followed by negative OLR and PAR excursions during early February (Fig. 6). This pattern was broadly similar to 1998, when peak temperatures were followed a few days later with anomalously low OLR. Additionally, there was a second SST peak during 2004 in mid- to late-February that aligned closely with positive OLR and PAR anomalies (Fig. 6).

In contrast to the undetectable differences of OLR and wind conditions between 2004 and bleaching years, the oceanographic setting in JFM of 2004 diverged clearly from bleaching years (1998 and 2002, as SODA does not include 2016 and 2017). Both 1998 and 2002 were characterized by an intensified EAC offshore of the central and southern GBR, associated with anomalously high SSH in the Coral Sea and low SSH on the GBR (Fig. 9). Conversely, the EAC was anomalously weak in 2004, and SSH in the Coral Sea was anomalously low while SSH on the GBR was near average (Fig. 9). Linked to the EAC and SSH fields, the depth-profile of isotherms during 1998 and 2002 tilted upwards from the Coral Sea towards the GBR shelf, whereas the isotherms were nearly flat across the Coral Sea in 2004 (Fig. 9). "Moderate" bleaching also occurred during 1982 and 1987 on the central GBR (*Hughes et al., 2018a*), and we find similar oceanographic settings during these 2 years as the severe bleaching events of 1998 and 2002 (Fig. S6).

The wind stress curl field in SODA displayed a broadly similar temporal pattern among 2004 and bleaching years as the oceanographic setting described above (Fig. 10). Wind stress curl was consistent with positive vertical Ekman transport (upwelling-favorable) across most of the GBR during JFM of 1998 and 2002. The opposite occurred during JFM of 2004, with winds stress curl across most of the GBR consistent with negative vertical Ekman transport (downwelling-favorable).

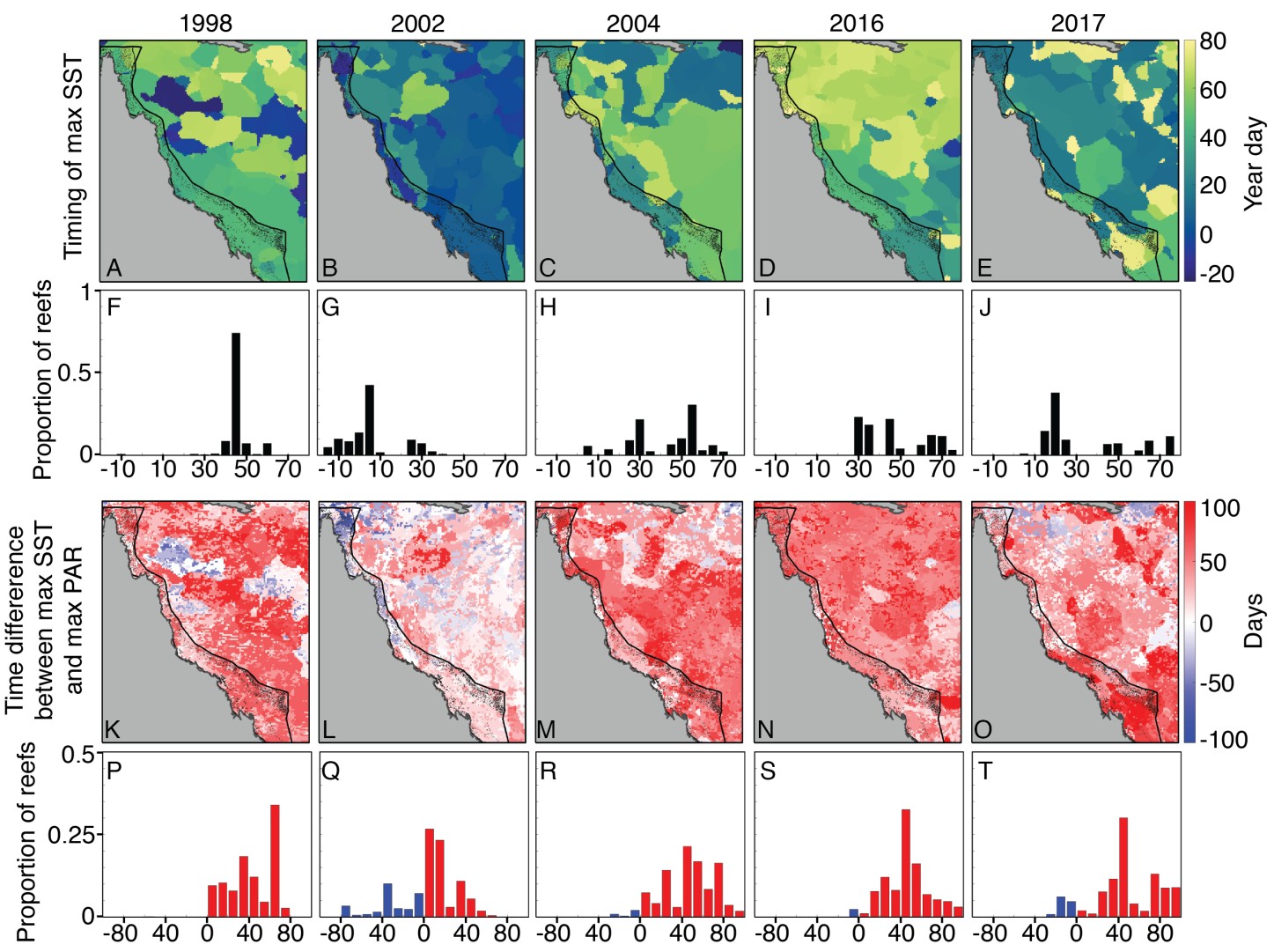

**Figure 8 Seasonal timing of SST maxima and their difference from PAR maxima.** A–E show the year day of maximum SST during 2004 and bleaching years (negative days indicate December of the preceding year), using OI-SSTv2 data. Histograms (F–J) with black bars show the proportion of GBR reefs with SST maximum occurring at various year days. K–O maps show the difference in days between maximum SST and maximum PAR, and the histograms in P–T show the corresponding proportions of GBR reefs. Red (blue) indicates that maximum SST occurred after (before) maximum PAR. SST maxima occurred relatively early in 2002, but the timing of maximum SST during 2004 is broadly similar to that of 1998, 2016, and 2017.

## DISCUSSION

### The history of heat stress on the GBR

Satellite-derived SSTs suggest that the austral summer of 2004 was one of the warmest on record across the GBR (Fig. 1). Yet, critically, the intensity and spatial distribution of heat stress varies drastically among four different gridded SST products (Fig. 2; Fig. S3). CRW, which is commonly used in assessments of GBR bleaching (*Liu, Strong & Skirving, 2003*; *Hughes et al., 2018b*), shows high DHWs offshore of the GBR and in the Coral Sea, but with the GBR shelf representing a distinct boundary to these high DHWs such that only 12% of GBR reefs experienced >4 °C-weeks during 2004 (Fig. 3). In stark

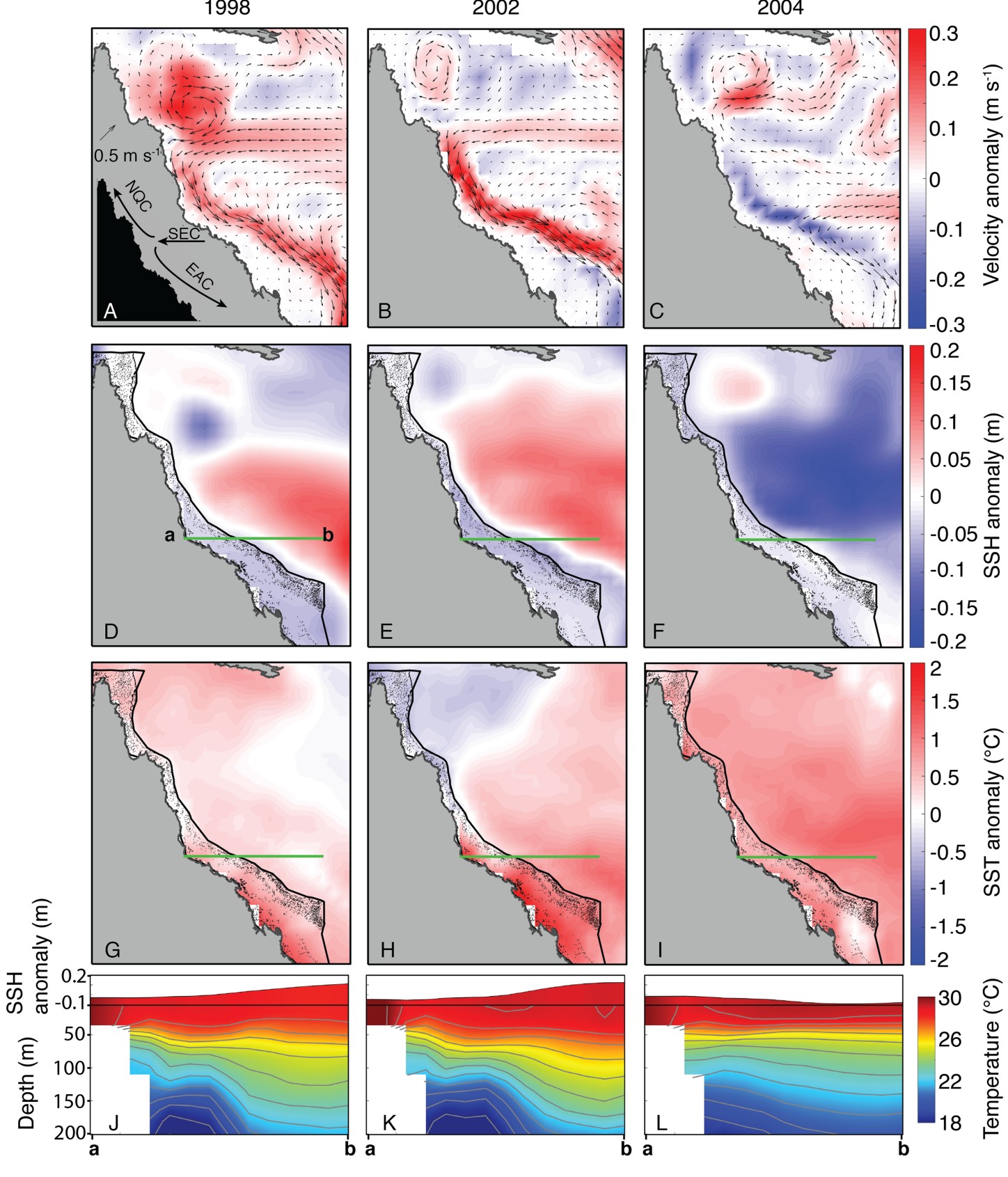

**Figure 9 Sea surface currents, height, and temperature anomalies for January, February, and March of bleaching years (1998, 2002, 2004) included in SODA.** In the velocity plots (A–C), arrows indicate the absolute current velocity, and colors represent the anomaly of current velocity relative to the climatological mean (1980–2015). In the SSH (D–F) and SST (G–I) plots, colors likewise indicate anomalies relative to the climatological mean. Green lines indicate the location of the temperature and SSH cross-sections shown in J–L (location of cross-section corresponds to the main upwelling intrusion passages; see *Benthuysen et al., 2016*). Inset image in (A) shows the general location of each current, although the location of the bifurcation point varies over time. EAC, East Australian Current; NQC, North Queensland Current; SEC, South Equatorial Current.

contrast, OI-SSTv1, OI-SSTv2, and CMC all suggest that high DHWs extended from the Coral Sea across the GBR shelf, with 81%, 67%, and 79% of reefs experiencing >4 °C-weeks, respectively, (Fig. 3). This difference is important because DHWs exceeding 4 °C-weeks have been observed to trigger a stress response that leads to bleaching (*Heron et al., 2016*; *Hughes et al., 2017*, *2018b*). Although the use of a single MMM climatology for all our analyses could affect the absolute values of DHW, the most important result is the comparison of 2004 to other years within each product. While CRW paints 2004 as a minor heat stress event compared to 2016–2017, OI-SSTv1, OI-SSTv2, and CMC all indicate that heat stress in 2004 was sufficient to initiate widespread bleaching, and that similar or greater numbers of reefs experienced >4 °C-weeks in 2004 compared to the severe bleaching years of 1998, 2002, 2016, and 2017 (Fig. 3).

The key differences in heat stress histories among the satellite SST products demonstrate the need for validation with in situ temperature loggers (*Pan et al., 2017*; *Fiedler et al., 2019*). Using all available temperature loggers from the AIMS database (*Australian Institute of Marine Science (AIMS), 2018*), we show that OI-SSTv2 provides the most accurate representation, on average, of reef-water temperatures (Table 1) and heat stress (Table 2) during austral summer 2004 on the GBR. While the accuracy of OI-SSTv2 remained relatively consistent across bleaching years, CRW accuracy changed substantially over time such that temperatures during 1998, 2002, and 2004 were underestimated compared to 2016 and 2017. These findings have important ramifications for interpreting the history of heat stress on the GBR. First, according to CRW, heat stress in 2016 and 2017 far exceeded that of previous bleaching years and 2004. However, although CRW or CMC provided the most accurate representation of temperatures in both 2016 and 2017, temperatures during earlier years were likely underestimated by CRW, potentially due to less coverage and lower-quality satellite data (*Liu et al., 2014*). In particular, either 2002 or 2004 was the year of highest DHW on at least half of the GBR according to OI-SSTv2, which was the most accurate SST product during these 2 years (Fig. 3). Second, our analysis indicates that 2004 heat stress reached sufficient levels to spark mass coral bleaching (Fig. 3 histograms). This makes 2004 enigmatic because mass bleaching was not reported.

While OI-SSTv2 was overall the most accurate satellite SST product in austral summer 2004, it is important to recognize that substantial uncertainties remain. For instance, OI-SSTv2 performed best for capturing maximum annual DHW for the Coral Sea, northern GBR, and many inshore reefs, but CRW performed best for a cluster of reefs in the central GBR (Fig. 5). Thus, we cannot conclude that OI-SSTv2 is a true representation of 2004 temperatures across the entire GBR, but rather that OI-SSTv2 provides a

better representation of 2004 heat stress on average across the GBR than CRW. Though CRW consistently underestimates temperatures and heat stress during 2004 (Fig. 4; Tables 1 and 2), OI-SSTv2 is an approximately even balance between overestimation in some locations and underestimation in others (Figs. 4 and 5). Thus, while these issues remain important, we can still conclude that temperatures on many reefs in 2004 reached levels that were sufficient to cause severe coral bleaching in other years. In the following sections, we explore potential reasons why widespread bleaching did not occur on the GBR in 2004. We place particular emphasis on the effects of environmental factors besides temperature that can differentiate 2004 from mass bleaching events.

## Did bleaching occur but was not observed?

Widespread bleaching on the GBR during 2004 is conspicuously absent from all existing coral bleaching databases (*Donner et al., 2017*; *Hughes et al., 2018a*; *Oliver, Berkelmans & Eakin, 2018*). However, it is critical to recognize that reporting bias can exist in such databases (*Oliver, Berkelmans & Eakin, 2018*), and that an absence of bleaching reports should not necessarily be taken as an absence of bleaching. Nevertheless, the GBR is one of the most closely monitored reef systems in the world. It is regularly surveyed by government agencies and scientists, and visited by hundreds of thousands of tourists every year. In 2004, the AIMS long term monitoring program (LTMP) reported low-level bleaching (scattered bleaching of individual colonies with a maximum of 3% of corals bleaching) between January and May on 20 of 80 reefs surveyed (*Sweatman et al., 2005*; AIMS LTMP; see Fig. S7). Additionally, analyses of density banding in coral skeletal cores from the central and northern GBR captured the bleaching event in 2002 but did not show evidence of mass bleaching in 2004 (*Cantin & Lough, 2014*; *DeCarlo et al., 2019*). While it is impossible to conclude with complete certainty that bleaching on the GBR was not more severe in 2004, there is a low probability that sustained and widespread bleaching went unnoticed.

Meanwhile, benthic surveys of Lihou Reef in the central Coral Sea Marine Park reported 65% of hard corals were bleached in March 2004 (*Oxley et al., 2004*), commensurate with the high levels of thermal stress (DHW of 13.1 °C-weeks) to which they were exposed (see point in Fig. 2). Although the bleaching observations at Lihou, 300 km from the GBR, were the only direct bleaching reports in the Coral Sea at that time, they suggest a strong disconnect from the GBR, where many more direct observations are available but no extensive bleaching was reported.

One consideration is that weather conditions could, at times, hamper monitoring efforts and obscure detection of bleaching. Two major tropical cyclones affected the GBR and Coral Sea region during summer 2004. The first, cyclone Fritz, impacted the far northern GBR in mid-February (Fig. S8), but this region was characterized by the lowest heat stress during 2004, except for some of the outer-shelf reefs (Fig. 2). Thus, while cyclone Fritz may have played a role in reducing heat stress in the far northern GBR, it is unlikely that the storm prohibited detection of bleaching along the entire GBR. Tropical cyclone Grace formed in the Coral Sea offshore of the central GBR on 20 March before moving southeast toward New Caledonia (Fig. S8). It passed directly over Lihou Reef 3 days after surveys reported bleaching (*Oxley et al., 2004*), and while it may have prevented

extensive coral mortality, it clearly did not prevent bleaching. Although cyclone Grace never directly struck the GBR, it is possible that this cyclone discouraged or prevented observers from visiting parts of the reef at that time. However, the cyclone formed approximately 2 weeks after maximum DHWs were reached in early-mid March and its core remained hundreds of kilometers away from the GBR. Therefore, we find it possible, yet very unlikely, that widespread bleaching did occur but was not observed.

## Did irradiance, winds, or currents mitigate bleaching?

Coral bleaching susceptibility is known to depend on a variety of other factors besides temperature, including light and nutrients (*Brown, 1997*; *Lesser, 1997*; *Fitt et al., 2001*; *Wooldridge, 2009, in press*; *Skirving et al., 2017*). Therefore, it is possible that bleaching did not occur in 2004 despite the high heat stress because of anomalous cloud cover, winds, or currents. Cloudiness (as inferred from OLR), and wind speed were relatively low in austral summers of 2004 and bleaching years (Fig. 6). Likewise, PAR was high across the entire GBR in 2002, and high in the warmest sections of the GBR in 2004 (central to southern), 2016 (northern), and 2017 (central to northern) (Fig. 7). The clear and calm conditions likely contributed, at least in part, to the warm reef-waters during these years by enhancing air-sea heat flux (*Schiller et al., 2009*; *Davis et al., 2011*; *DeCarlo et al., 2017*; *Benthuysen et al., 2018*). Beyond their influence on temperature, cloud and wind conditions could enhance the light reaching corals, thereby increasing their sensitivity to heat stress (*Payne, 1972*; *Stramska & Dickey, 1998*; *Davis et al., 2011*). In the Keppel Islands of the far southern GBR, the absence of bleaching despite anomalously high temperatures in 2004 was attributed to relatively low light stress during that summer (*Skirving et al., 2017*). Indeed, our analysis of MODIS PAR data shows the Keppel Islands were exposed to PAR levels comparable to the climatological mean (Fig. 7). However, north of the Keppel Islands, in much of the central to southern GBR, PAR anomalies were higher at this time. Furthermore, in our analysis of OLR and wind-speed across the entire GBR, austral summer 2004 was not detectably cloudier or windier than all bleaching years, making it unlikely that differences in irradiance can explain the large-scale absence of bleaching in 2004. The seasonal timing of maximum SST has also been considered as a factor affecting bleaching sensitivity (*Bahr, Jokiel & Rodgers, 2015*). However, 2004 is not clearly differentiated from all bleaching years based on the timing of maximum SST or the time difference between maximum SST and maximum PAR (Fig. 8).

Nutrient levels can also modulate coral susceptibility to heat stress (*Wooldridge, 2009, in press*; *Cunning & Baker, 2012*; *Wiedenmann et al., 2013*; *D'Angelo & Wiedenmann, 2014*; *Vega Thurber et al., 2014*; *Baker et al., 2018*; *Wang et al., 2018*). Excess nutrients increase the concentrations of zooxanthellae within coral tissues (*Marubini & Davies, 1996*; *Wooldridge, in press*), which can become detrimental when temperatures exceed normal summertime levels and toxic levels of oxygen produced by the zooxanthellae lead to the breakdown of the symbiosis (*Lesser, 1997*; *Fitt et al., 2001*; *Cunning & Baker, 2012*). A key process affecting nutrient levels on the GBR is upwelling of Coral Sea water onto the shelf through passages, primarily located in the central GBR (*Andrews & Gentien, 1982*; *Wolanski & Pickard, 1983*; *Furnas & Mitchell, 1996*; *Berkelmans, Weeks &*

*Steinberga, 2010*; *Benthuysen et al., 2016*). The deeper waters that upwell onto the shelf are nutrient-rich, and once they reach the shelf they are retained for weeks to months as they spread for hundreds of kilometers within the GBR lagoon (*Andutta, Ridd & Wolanski, 2013*; *Benthuysen et al., 2016*). Although these waters rarely reach the sea surface directly, they can represent an important source of nutrients into an otherwise largely oligotrophic system (*Furnas & Mitchell, 1996*; *Berkelmans, Weeks & Steinberga, 2010*). Upwelling through the reef passages and onto the shelf is strongly influenced by the strength of the EAC offshore of the GBR (*Steinberg, 2007*; *Berkelmans, Weeks & Steinberga, 2010*; *Benthuysen et al., 2016*). A faster-flowing EAC is generally associated with an anomalously steep SSH gradient sloping downward toward the shelf, and isotherms tilting upward toward the shelf (*Steinberg, 2007*; *Benthuysen et al., 2016*). Our analysis of SODA indicates that this was precisely the oceanographic setting during the 1998 and 2002 severe bleaching events (Fig. 9), and during the 1982 and 1987 moderate bleaching events (based on definitions of *Hughes et al., 2018a*; although 1982 was arguably a severe event based on our interpretation of *Oliver, 1985*) (see Fig. S6). Indeed, detailed analysis of in situ temperature logger data showed that upwelling occurred more frequently than normal during summers 1998 and 2002 (*Berkelmans, Weeks & Steinberga, 2010*), potentially contributing to widespread bleaching by increasing nutrient levels along the GBR (*Wooldridge, 2009*, *in press*). Critically, the opposite pattern persisted through the austral summer of 2004. The EAC was sluggish, SSH in the Coral Sea was anomalously low, and isotherms remained nearly flat (Fig. 9), conditions which inhibit upwelling onto the GBR shelf. Therefore, the slow EAC in 2004 may have played a role in sparing the GBR from widespread bleaching.

The potential role of the EAC in modulating bleaching sensitivity on the GBR is distinct from other current systems that may reduce surface temperatures. For example, the Equatorial Undercurrent collides with coral reefs in the equatorial Pacific Ocean, upwelling cold water to the surface and directly mitigating thermal stress on resident corals (*Gove, Merrifield & Brainard, 2006*; *Karnauskas & Cohen, 2012*; *Barkley et al., 2018*). Conversely, the EAC does not appear to mitigate bleaching by reducing heat stress, but rather it can potentially enhance bleaching sensitivity by adding nutrients into the GBR lagoon system without substantially affecting surface temperatures (*Berkelmans, Weeks & Steinberga, 2010*). Furthermore, the EAC typically only affects the central and southern GBR, whereas the North Queensland Current (NQC) flows toward the equator adjacent to the northern GBR (*Schiller et al., 2009*; *Andutta, Ridd & Wolanski, 2013*). In fact, slowdowns in the NQC can result in localized heating along the northern GBR, which likely contributed to the bleaching in this region during 2016 (*Wolanski et al., 2017*). Critically, the geographic footprints of >4 °C-weeks and the anomalously slow EAC mostly overlap across much of the central to southern GBR during summer 2004 (Figs. 2 and 9), which were the areas most intensely monitored at this time (Fig. S7). Some of the outer-shelf reefs of the far northern GBR also experienced >4 °C-weeks in the same area where the NQC was anomalously slow (Figs. 2 and 9), although this area was not surveyed as extensively during the summer of 2004 (Fig. S7). Thus, our interpretation of the EAC influencing the absence of bleaching in 2004 applies primarily to the central and

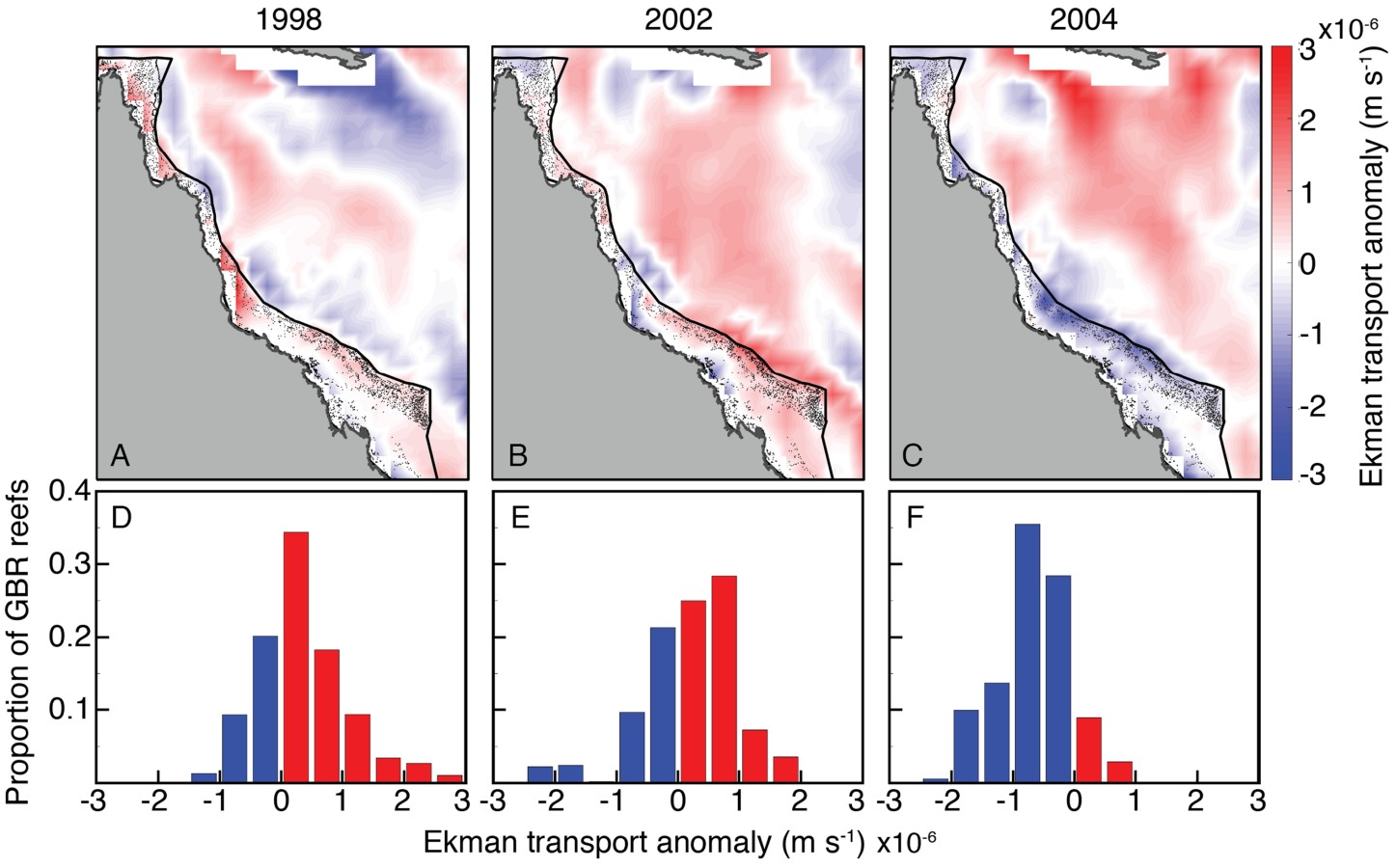

**Figure 10 Ekman vertical transport anomalies for January, February, and March of bleaching years (1998, 2002, 2004) included in SODA.** Red (blue) indicate anomalously upwelling (downwelling) favorable wind stress curl in both the maps (A–C) and histograms (D–F).

southern GBR, with the outer-shelf reefs of the far northern sector being more likely to have experienced an unnoticed bleaching event.

Another related process capable of affecting upwelling is wind stress curl (*Enriquez et al., 1995*; *Chelton et al., 2004*). Most of the GBR was exposed to upwelling-favorable wind stress curl in austral summers of both 1998 and 2002, but anomalous conditions that favored downwelling during 2004 (Fig. 10). Thus, wind stress curl may have further enhanced the upwelling that occurred during bleaching years while dampening upwelling during 2004, when heat stress was high but bleaching did not occur.

## Did corals acclimatize after the 1998 and 2002 bleaching events?

Acclimatization is the adjustment of an organism to variations in its environment, such that it can tolerate a wider range or a different set of conditions. Corals exhibiting lower bleaching susceptibility following previous exposure to heat stress is often taken as evidence of acclimatization to higher temperature regimes (*Berkelmans & Willis, 1999*; *Guest et al., 2012*; *Pratchett et al., 2013*; *Coles et al., 2018*; *Gintert et al., 2018*; *Palmer, 2018*; *DeCarlo et al., 2019*). This can occur through a variety of processes such as hosting

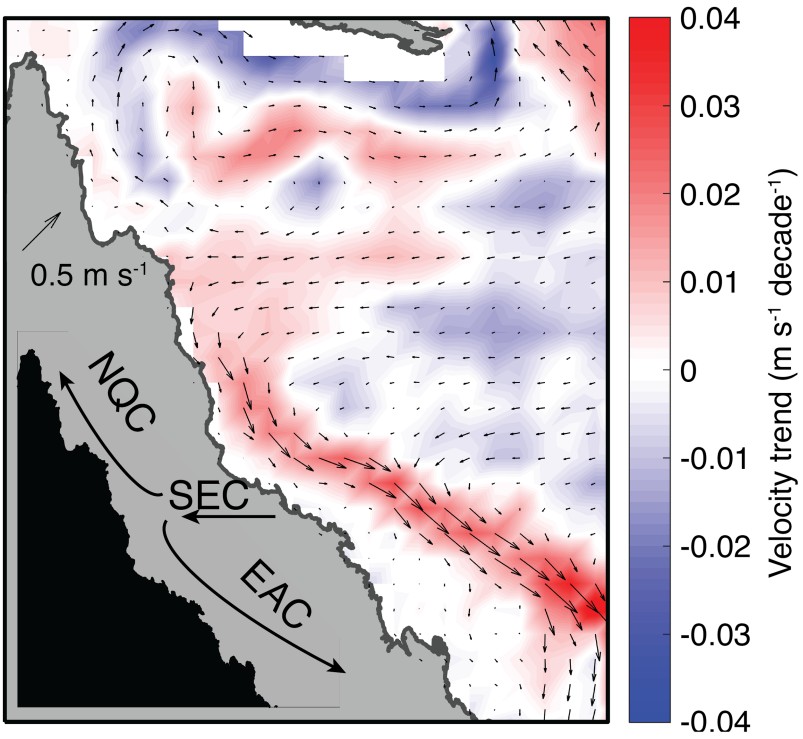

**Figure 11 Trends in near-surface (0–25 m) velocity during January, February, and March in SODA (1980–2015).** Colors indicate the rate of velocity change, and arrows indicate the climatological mean velocity field.

new zooxanthellae populations or immune system responses (*Buddemeier & Fautin, 1993*; *Putnam et al., 2017*; *Palmer, 2018*). Additionally, corals on some shallow reefs with relatively high daily temperature ranges may gain tolerance to heat stress (*Safaie et al., 2018*). Coral communities may also become less sensitive to heat stress if previous bleaching events eliminate the most susceptible species or individuals (*Vargas-Ángel et al., 2001*; *Harrison et al., 2018*; *Hughes et al., 2019*). Nevertheless, if we were to consider bleaching as solely a response to temperature, then the absence of bleaching on the GBR in 2004 would be a strong indication of acclimatization because mass bleaching occurred in 1998 and 2002 under similar or less heat stress. However, this logic may not hold if other factors modulate the bleaching susceptibility to temperature. For instance, if lower nutrient levels in 2004 reduced bleaching sensitivity, then acclimatization may not have been necessary. Rather, conditions during 2004 may simply have been less stressful if nutrient levels were lower, even though temperatures were exceptionally high. Therefore, acclimatization remains a potential factor in contributing to the absence of bleaching in 2004, but we cannot conclude with certainty that it occurred to a meaningful extent.

## CONCLUSION

The austral summer of 2004 on the GBR is enigmatic because it was one of the warmest summers on record, yet only very low levels (<3%) of bleaching were recorded. Our analyses of satellite SST products, validated against a suite of in situ temperature loggers,

indicates that temperatures were sufficiently high in 2004 to trigger widespread coral bleaching, based on observed bleaching thresholds in other years. Ocean currents–but not winds, cloud cover, or temperature–differentiate 2004 from bleaching years. Specifically, a slowdown of the EAC during austral summer 2004 established conditions unfavorable for upwelling of nutrient-rich water on the GBR shelf. We infer that nutrient levels may play a role in modulating coral bleaching susceptibility on the GBR, and that changes in the strength of the EAC can potentially either mitigate or enhance bleaching. Since 1980, the EAC has accelerated (see also *Hill et al., 2008*; *Suthers et al., 2011*) and the NQC has decelerated (Fig. 11). If the strength of the EAC continues to increase as expected due to the "spin-up" of the SEC (*Oliver & Holbrook, 2014*; *Roemmich et al., 2016*), the GBR could become even more prone to mass bleaching events as global warming drives more frequent marine heatwaves.

Understanding the role of oceanography and environmental variables in corals' responses to heat stress requires additional testing in laboratory and natural settings, as well as long-term in situ measurements of irradiance, winds, currents, and nutrients. We suggest that continuous monitoring of a greater suite of environmental variables in coral reef ecosystems is necessary to better quantify bleaching thresholds and corals' adaptive capacity in an era of rapid environmental change. Nevertheless, our findings clearly highlight the dynamic nature of coral bleaching thresholds during marine heatwaves. Other examples of an absence of coral bleaching despite high temperature stress exist in well-monitored areas, for example during 1968 in Hawai'i (*Bahr, Jokiel & Rodgers, 2015*), and investigating such case studies can advance our understanding of coral heat tolerance.

## ACKNOWLEDGEMENTS

We thank the Australian Institute of Marine Science for providing bleaching reports from the Long Term Monitoring Program. We are grateful to the organizations that made the climate data used in this study publicly available, and to Kris Karnauskas (University of Colorado Boulder), Jessica Benthuysen (AIMS), and Mike Emslie (AIMS) for insightful discussions. We thank Dr. Keisha Bahr and an anonymous reviewer for their constructive comments.

### Funding

Hugo B. Harrison was supported by ARC Discovery Early Career Research Award (DE160101141). The funders had no role in study design, data collection and analysis, decision to publish, or preparation of the manuscript.

### Grant Disclosures

The following grant information was disclosed by the authors:
ARC Discovery Early Career Research Award: DE160101141.

## Competing Interests

The authors declare that they have no competing interests.

## Author Contributions

- Thomas M. DeCarlo conceived and designed the experiments, analyzed the data, prepared figures and/or tables, authored or reviewed drafts of the paper, approved the final draft.
- Hugo B. Harrison conceived and designed the experiments, authored or reviewed drafts of the paper, approved the final draft.

## Data Availability

Raw data and code are available at: https://codeocean.com/capsule/8487532/.

## Supplemental Information

Supplemental information for this article can be found online at http://dx.doi.org/10.7717/peerj.7473#supplemental-information.

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
