# Peer review of "An enigmatic decoupling between heat stress and coral bleaching on the Great Barrier Reef"

_PeerJ, doi:10.7717/peerj.7473_

## Round 0.1 · original submission · Major Revisions

The paper has been evaluated by two reviewers. I agree with Reviewer 1, currently the manuscript needs a major revision, clarifications of the figures and answering and interpreting Upwelling nutrients and the statements on coral bleaching need to be carefully answered within the discussion. Please also correct all minor and other major comments by both reviewers. I would be happy to see a revised version of this manuscript with a clear rebuttal letter.
Good luck.

Reviewer 1 ·

Basic reporting

Well written paper.

Experimental design

High standard.

Validity of the findings

The conclusions are given tentatively. I don't agree entirely with their conclusions, but I think the key point is the results (or more correctly analysis of the results) are soundly undertaken.

Additional comments

This paper poses an excellent question (why didn’t 2004 bleach on GBR), and uses a broad range of available data to answer it. The paper is very well written. The tentative conclusion of the paper is that current driven upwelling, a function of the alongshore velocity of the East Australian Current, is the explanation. While they are looking at important processes, I am not convinced the EAC is the dominant reason (I suspected it is solar radiation, as per point 3). They could also have made a better critique of their conclusion by considering spatially-resolved differences in the bleaching, or lack thereof, in 2004. Nonetheless, they do not over state their findings, and give detailed considerations to an important question. So I conclude this is a paper that should be published, and will provide new estimates on one component of an important, complex problem. But the below comments should be addressed.
Major comments.
1. I found the figures of the whole GBR with the corals as black dots to be frustrating. What we really want to see is the temperature over the reefs, but instead this is blacked out, and we a drawn to regional scale processes such as offshelf Coral Sea water. I think these figures should be redrafted as scatter plots, with reef points only given and coloured by temperature.
2. For the discussion. There are papers on the decadal oscillations of the EAC that would put your arguments in perspective (search Hill, Ridgway Cetina-Heredia). As I understand it, but the authors should check, the EAC (at least south of 30 S) was at a maximum strength in around 2005, and a minimum around 2009. I suggest that you look into this using the model you have (SODA), and for your relevant latitude. If the oscillations show a decadal signal, then you can make estimates of when the EAC influence on bleaching will be greatest. What does the next decade hold for EAC-influences on bleaching?
3. When I look at figure 6, OLR, I would argue the main reason that corals didn’t bleach is that the length of time with anomalously high OLR was shorter for 2004 than the other years. Could you add a third row to this figure of temperature? Corals bleach because of light absorbed at elevated temperatures. How well do the excess light and elevated temperatures line up in each year?
4. Upwelling nutrients, if limiting, will (1) increase zooxanthellae (thus increasing oxidative stress per coral host), (2) reduce nutrient limitation, thus potentially shifting to light limitation, and encouraging more pigment synthesis that could drive more oxidative stress per symbiont) OR (3) increase the activity of Rubisco enzyme, therefore decreasing oxidative stress.
5. Upwelled nutrients will not spread evenly over the reef. The argument for the role of the EAC would be much strengthened if it is possible to show that the EAC has a greater impact on preventing bleaching where it has the greatest influence.
6. North of the separation point, the alongshore flow is northward, which is downwelling favourable. So a weaker SEC would shift the corals north of the separation point to more impacted by upwelled nutrients, and therefore more likely to bleach. The counter argument would be that north of the separation point upwelling is driven mostly by tides, with current-driven bottom boundary layer dynamics less important. The differing effect of the SEC intensity north and south of the separation point should be considered.
Minor comments.
1. What does MODIS do with PAR data during cloudy periods? I suspect a satellite weather products might do better than MODIS PAR – but this will no substantially change the story.
2. L210 ‘South Equatorial Current (SEC) colliding with the Australian continental shelf at around 15 S’
3. L214 ’x’ and ‘y’
4. Fig. 4 – choose a different colour for Hotspot days (loggers only).

·

Basic reporting

The article is well written, unambiguous, and technically clear. Authors provide sufficient background and extensive rationale for research. They highlight the importance of understanding the variety of environmental drivers (besides temperature) the can contribute to coral bleaching. The structure of the manuscript is well organized and easy to follow. I appreciated the sufficient description of their methods and rationale for use of SST derived products.

Experimental design

Methods are describe with sufficient detail. Authors provide detailed information for replicating work. Authors provide thorough evidence of error and standardization of each method.

Line 124: Can authors provide evidence of accuracy for direct comparison of overlaying SST estimations? Is there previous evidence of linear interpolation method to create a pixel-by-pixel comparison of the SST products?

Validity of the findings

Authors highlight importance of key differences in satellite SST products and the need for validation with in situ loggers. I would like to see a text including their recommendation on which SST products to use in future coral bleaching work.

Clarification is needed regarding the reported 64% bleaching that occurred in the Coral Sea Marine Park. Please make a clear distinction between your claims of <3% bleaching and the bleaching that was recorded in the Coral Sea Marine Park. Currently, the manuscript is overlooking the extent of bleaching that did occur in the Coral Sea Marine Park. How did SST products relate to the observe bleaching there?

Additional comments

Very well written manuscript that highlights the importance of monitoring a suite of environmental variables in coral reef ecosystems to better quantify bleaching thresholds. This information is crucial in understanding how to move coral bleaching research forward.

---

## Round 0.2 · accepted · Accept

Congratulations, happy to see this paper out.

Reviewer 1 ·

Basic reporting

No further comment.

Experimental design

No further comment.

Validity of the findings

No further comment.

Additional comments

I am happy with the authors' modifications on what was already a nice piece of work.